# A Review of the Stress Resistance, Molecular Breeding, Health Benefits, Potential Food Products, and Ecological Value of *Castanea mollissima*

**DOI:** 10.3390/plants11162111

**Published:** 2022-08-14

**Authors:** Yanpeng Wang, Cuiyu Liu, Zhou Fang, Qiang Wu, Yang Xu, Bangchu Gong, Xibing Jiang, Junsheng Lai, Jingen Fan

**Affiliations:** 1Research Institute of Subtropical Forestry, Chinese Academy of Forestry, Hangzhou 311400, China; 2Qingyuan Bureau of Natural Resources and Planning, Lishui 323800, China; 3State Key Laboratory of Tree Genetics and Breeding, Chinese Academy of Forestry, Beijing 100091, China; 4Lanxi City Nursery of Zhejiang Provence, Lanxi 321100, China

**Keywords:** *Castanea* spp., *Phytophthora cinnamomi*, simple sequence repeat, germplasm resources

## Abstract

Chestnut (*Castanea* spp., *Fagaceae* family) is an economically and ecologically valuable species. The main goals of chestnut production vary among species and countries and depend on the ecological characteristics of orchards, agronomic management, and the architecture of chestnut trees. Here, we review recent research on chestnut trees, including the effects of fungal diseases (*Cryphonectria parasitica* and *Phytophthora cinnamomi*) and insect pests (*Dryocosmus kuriphilus* Yasumatsu), molecular markers for breeding, ecological effects, endophytic fungi, and extracts with human health benefits. We also review research on chestnut in the food science field, technological improvements, the soil and fertilizer used for chestnut production, and the postharvest biology of chestnut. We noted differences in the factors affecting chestnut production among regions, including China, the Americas, and Europe, especially in the causal agents of disease and pests. For example, there is a major difference in the resistance of chestnut to *C. parasitica* in Asian, European, and American countries. Our review provides new insights into the integrated disease and pest management of chestnut trees in China. We hope that this review will foster collaboration among regions and help to clarify differences in the direction of breeding efforts among countries.

## 1. Introduction 

Chestnut (*Castanea* spp., *Fagaceae* family) is an economically important tree in the wood processing industry that occurs in Asia, Europe, Africa, and the Americas [1,2,3]. There are four most important and cultivated chestnut varieties: *C. molissima* (Chinese chestnut) and *C. crenata* (Japanese chestnut) are distributed in Asia; *C. sativa* is distributed in continental Europe (European chestnut); and *Castanea*
*dentata* is distributed in North America (American chestnut) [4]. Chestnuts are a source of nuts and industrial raw materials, including wood, that can be used as firewood, as well as to build timber and barrels for winemaking (Figure 1). In recent years, extracts from chestnut shells (e.g., tannins, polyphenols, and polysaccharides), female flower, and spring buds have been applied in the medical, pharmaceutical, and healthcare fields [5,6,7,8] (Figure 1). Discarded chestnut shells, inner shells, burs, and leaves have been re-utilized as biomass and catalyst material [9,10]. The important economic and environmental roles of chestnut underlie its high value to ecosystems and agroforestry systems [11,12,13,14] (Figure 1).

Chestnut plantations in Europe and North America are vulnerable to diseases caused by *Cryphonectria parasitica* and *Phytophthora cinnamomi* and damage from pests, such as *Dryocosmus kuriphilus* Yasumatsu (*Hymenoptera*, *Cynipidae*) [15,16,17,18,19]. Diseases and pests pose a major threat to the production of chestnut plantations and related ecological systems (Figure 2). Nevertheless, global chestnut production has continually increased over the past 15 years, and this global increase is mainly driven by increases in chestnut production in China, which contributes nearly 90% of the world’s chestnut production [20]. Increases in chestnut production in China have largely been driven by government policies, scientific fertilization and planting technology, the development of new varieties and agricultural tools, and increases in the skills of farmers.

From 2019 to 2022, we conducted surveys of the planting technology, species of pests and disease, soil nutrients, and fertilization conditions of Chinese chestnut in Zhejiang and Hubei Provinces, and this survey was supported by the Research Institute of Subtropical Forestry, Chinese Academy of Forestry. Our survey revealed that the main focus of current research on chestnut varies among Asian, American, and European countries. Although blight and ink diseases are widespread in China and Japan, chestnut is naturally resistant to the agents causing these diseases, and this has prevented chestnut populations from becoming extirpated. However, American and European chestnuts do not possess natural resistance to the agents causing these diseases, and this has resulted in the complete extirpation of chestnut populations when nursery stocks were introduced to the Americas and Europe from China and Japan in the late 1800s [20]. Chestnut gall wasp is the main perennial pest of chestnut trees in Europe and the Americas; however, *Conogethes punctiferalis* (Guenée) is the main pest species that currently affects chestnut trees in Hubei Province, China (data not shown). In some chestnut orchards in Hubei Province and Zhejiang Province, the chestnut trees are either fertilized by chemical and organic fertilizers or not fertilized. In Europe (Italy), the organic matter content is increased through the addition of sheep and cattle; alternatively, chestnut trees are fertilized with 10–15 tons/ha/year manure [3]. Breeders in America and Europe have focused on the use of molecular markers to detect resistant genes against fungal pathogens, and the quality of the nuts of chestnut trees has been a main focus of research on chestnut in China. These differences in the focus of research on a global scale require consideration for ensuring the health of nursery stocks in different regions. As the production and global importance of chestnut trees in China are growing, more academic collaboration among regions is needed to solve the problems associated with chestnut production.

## 2. Main Pathogens and Insect Pests and Their Monitoring

### 2.1. Disease and Pest Species

Chestnut blight is caused by *C.*
*parasitica*, which is a fungus that specifically infects stem tissue and eventually kills the tree. It has become the most economically significant disease of chestnut trees and has resulted in major yield reductions in North America and Europe. The adverse effects of blight fungus disease occur in the last growth ring next to the vascular cambium, where symptoms of deterioration can be observed [21]. The leaves of European chestnut (*Castanea*
*sativa* Mill.) have the lowest photosynthetic capacity but the highest antioxidant enzyme activities in response to blight fungus disease when compared to other chestnut species [16].

The fungus *P. cinnamomi* is another pathogen that causes ink disease and poses a threat to the health of chestnut forests. *P. cambivora*, *P. citricola,* and *P. cactorum* are the main *Phytophthora* species that cause ink disease in chestnut trees, but *P. cambivora* is the most aggressive to *Castanea sativa* [22]. The Chinese chestnut (*Castanea*
*mollissima*) and Japanese chestnut (*Castanea*
*crenata*) are innately resistant to *P. cinnamomi*, an oomycete that causes ink disease. However, European chestnut has little to no disease resistance to this pathogen; consequently, this pathogen can induce substantial losses in European chestnut forests, orchards, and nurseries. Significant changes to the proteome, the accumulation of proline, and an increase in the content of salicylic acid, methyl jasmonate, and sugars mediate the defense response to infection by *P. cinnamomi* [17]. Up-regulation of the expression of the plant susceptibility genes *pmr4* and *dmr6* negatively regulates the SA pathway, and negative regulation of the expression of defense genes increases susceptibility to infection by *C. parasitica* and *P. cinnamomi* [18].

Chestnut rot can have a significant negative effect on the quality and yield of chestnut products. Fungal isolates causing brown rot are homologous with *Gnomoniopsis* sp., *Gnomoniopsis Castanea*, and *Gnomoniopsis smithogilvyi*, and brown rot disease outbreaks can also appear following several periods of drought [22]. *Gnomoniopsis Castaneae* was first reported in Greece; it is the causal agent of nut rot and can reduce yields by approximately 20~30% [23].

The Asian chestnut gall wasp *Dryocosmus kuriphilus* Yasumatsu (*D. kuriphilus*, *Hymenoptera*, *Cynipidae*) is one of the most economically significant pests of chestnut, and it can severely reduce yields of nuts and timber [15,24]. Losses as high as 80% have been observed when the number of galls of *D. kuriphilus* exceeds six per 50-cm twig [14,25]. The development of resistant chestnut trees is an effective approach for reducing the number of pest eggs and the number of larvae that complete development, including the number and size of the galls of *D. kuriphilus* [26]. Biological control via the parasitoid *Torymus sinensis* is an effective approach for controlling *D. kuriphilus* in Portugal, and a mathematical model has shown that this biological control approach is efficient in small and homogeneous orchards [27].

### 2.2. Disease and Pest Monitoring

Comprehensive measures have been taken to reduce the spread of chestnut disease caused by different pathogens to ensure the health of forests and ecological security. Due to the high economic and ecological value of chestnuts, significant efforts have been made to combat pathogens over the past century. Recently, genomics approaches have been used to identify defense-related genes in *C. parasitica*. Construction of a linkage map and identification of quantitative trait loci (QTL) for resistance to *P. cinnamomi* of backcross families derived from crosses of American chestnuts with Chinese chestnuts have shown that their hybrid progeny has a few dominant QTL and possess quantitatively inherited partial resistance via multiple small-effect QTL [28]. A set of simple sequence repeat (SSR) markers from expressed sequence tags (ESTs) among European, American, Japanese, and Chinese chestnut species revealed a high interspecific transferability rate of EST-SSRs that facilitates the evolution of resistance to *P. cinnamomi* [29]. Transcriptome analysis has revealed that differentially expressed genes between canker and healthy stem tissue are involved in cell wall biosynthesis, reactive oxygen species, and hormones [30]. Recovery is slower in the marron genotype when injected with the hypovirulent strain of *C. parasitica* compared with normal chestnut trees, which indicates that the recovery from blight disease is genotype-dependent [31]. Biological agents might be a valuable approach for resisting pathogens. For example, *Brevibacterium frigoritolerans* DSM8801^T^ and *Arthrobacter parietis* LMG 22281^T^ are effective against both *C. parasitica* and *P. cinnamomi* in vitro [14]. Research has also shown that subsequent generations of ink-diseased chestnut trees show enhanced tolerance to *P. cinnamomi*, which might stem from the fact that the invasive pathogen induces resistance to pathogen stress [18]. Various approaches, including genetic engineering and conventional breeding, have been used to mitigate the negative effects of disease and insects. The reintroduction of resistant species is another efficient way to promote the recovery of forests in American and European countries [32].

### 2.3. Control of Diseases and Pests

To control the aforementioned diseases and pests, fruit growers in China have typically used chemical insecticides and fungicides because they are cheap and efficient. For example, mist sprays with insecticides and fungicides used in the ‘Resource Nursery of Chinese chestnut’ (Lanxi City, Zhejiang Provence, China) are highly effective for controlling damage caused by pathogens and pests. Previous studies over the past five years have revealed that the combination of Dithane M-45^®^ (80% WP, Dow AgroSciences^®^, Indiana, United States), Score^®^ (10% WG, Difenoconazole, Syngenta, Basel, Switzerland), and Cabrio^®^ (250 g/L EC, BASF^®^, Ludwigshafen, Germany) is the most effective for controlling infection by *P. cinnamomi* and *C. parasitica*. The combination of LORSBAN^®^ (Chlorpyrifos, 45% EC, Dow AgroSciences^®^, Indiana, United States) and MEOTHRIN^®^ (Fenpropathrin, 20% EC, Sumitomo Chemical(Shanghai) Co., LTD., Shanghai, China) is the most effective for controlling infection by *D. kuriphilus, D. punctiferalis*, and other lepidopteran pests.

Chemical control treatments also have several shortcomings, such as the high costs of applying them on larger trees, the killing of beneficial insects (e.g., *Torymus sinensis*), and environmental pollution. The only alternative to chemical, physical, and biological measures is the breeding of new genotypes with resistance to pathogens, which can decrease and weaken the pest population [25].

## 3. Iatrical Benefits and Any Other Functions

Glucan extracted from *Castanea*
*mollissima* fruit and its selenylation modification derivative have strong antiproliferative effects and could potentially be used as an antitumor drug [33]. Castanol B (1), a new phenol from a water-soluble extract of shells of *Castanea*
*mollissima*, suppresses the growth of hepatoma cells and induces cell apoptosis by reducing the levels of TLR4, IKKβ, and NF-κB [34]. CMP90 is a new water-soluble polysaccharide isolated from *Castanea*
*mollissima* Blume that inhibits the proliferation of HL-60 cells and the growth of S180 solid tumors [8]. Selenium nanoparticles decorated with 1,6-α-D-glucan (isolated from the fruits of *Castanea*
*mollissima* Blume) have an anti-proliferative effect on HeLa cells in vitro by inducing apoptosis and S phase arrest [32]. All extracts from the chestnut shell, inner shell, bur, and leaves exhibit antioxidant activity and inhibit the growth of all Gram-positive and two Gram-negative bacteria [35]. Ellagic acid and chestanin extracted from the burs of *Castanea sativa* Miller inhibit the growth of *Alternaria alternata* and *Fusarium solani* by inhibiting mycelial growth and spore germination [36]. It is also reported that chestnut male flower has both excellent antioxidant and antimicrobial properties [37,38] (Figure 1). Portuguese delicacy “pastel de nata” with added extracts of *Castanea*
*sativa* male flower had higher activity of reducing agents and radical scavengers two days after baking [39]. The leaf extract of *Castanea*
*sativa* Mill. reduces the level of malondialdehyde when human sperm samples are treated with H_2_O_2_ [40] and lowers serum cholesterol levels in mice [41]. A new report also showed that 30 g resistant starch had a significant protective role against non-colorectal cancers for patients, and chestnut is rich in resistant starch [42] (Figure 1). Extracts obtained from chestnut are hypoglycemic and show antioxidant activity, and this mitigates the negative effects of diabetes on the liver [43].

Bioactive compounds extracted from chestnut shells and bur, such as polyphenols, minerals, lignin, and vitamins, are widely used in the food and medicine industries. Polyphenols extracted from chestnut are considered to be beneficial bioactive compounds in food production and preservation [44,45]. *C. mollissima* kernel is rich in vitamin C and vitamin E, which are beneficial in humans’ daily diet in reducing the diseases of scurvy and cancer [14,46]. Various macroelements (such as K, P, Ca, and Mg) and microelements (such as Zn, Fe, and Cu) are contained in chestnut kernel, which are beneficial minerals to maintain health [47,48].

## 4. Industrial Production and Technology

### 4.1. Chestnut Products

Instead of burning waste chestnut shell, many studies have subjected them to biorefinery processes. The use of 5% (*w*/*v*) activated charcoal can yield 70.3% (*w*/*v*) phenolic compounds, and the phenol radical scavenging activity is higher in the first activated charcoal eluate than in crude chestnut shell hydrolyzate [9]. *Castanea mollissima* shell can be used in various reactions (e.g., propane dehydrogenation), and it shows high catalytic performance and is low in cost [49]. In vitro digestion has shown that the total phenol content and antioxidant activity are higher in the outer/inner skin of digested chestnut than in undigested chestnut, and this could be used as a source of raw materials for antioxidant-rich active substances [5].

### 4.2. Improvements in Industrial Technologies

High amounts of tannins are present in chestnut wood and bark, and these tannins are widely used in the wood, culinary, and medicinal industries; they are also used as a fuel, as an additive in animal fodder, and for tanning leather [6,37]. Experiments of the hydrothermal hydrolysis of sweet chestnut tannins have revealed that 250 °C for 30 min are the optimal conditions for maximizing the content of total tannins and phenols [6]. Optimized choline chloride-based deep eutectic solvents can be used to extract and recover ellagic acid from waste chestnut shells [7]. The content of flavonoids and tannins from male chestnut flowers is maximized when the following optimal conditions of ultrasonic-assisted extraction are used: 24 ± 3 min, 259 ± 16 W, and 51 ± 7% ethanol [50].

## 5. Food Science

### 5.1. Beneficial Extractions from Chestnut

The glycemic index is negatively correlated with the relative crystallinity, which indicates that the starch crystalline structure has a retardant effect on digestibility [51]. Various factors, such as the drying temperature and the order of long-range, short-range, and other molecular compounds in chestnut, affect the digestibility of starch [52]. Animals fed with ethylenediaminetetraacetic acid, a tannin-rich extract from chestnut, promote pig fattening by increasing the body weight, average daily weight gain, and the feed-to-gain ratio of pigs [53]. The addition of chestnut to the diet increases the intramuscular fat content of the biceps femoris muscle of pigs compared with commercial feed; a diet with chestnut also alters the content of some volatile compounds [54]. The organic acid composition is affected by hot air convective drying treatment, and the effect depends on the chestnut variety [55]. Soy protein isolate films containing chestnut bur extract might provide an effective packaging and preservation method given that antioxidant activity is increased in films with chestnut bur extract [56]. A blend of chestnut and wheat flour or rice flour can be used to generate a high-quality paste [57].

According to different authors, *C. mollissima* bur and inner shell are remarkable sources of bioactive ingredients. *C. mollissima* bur is rich in phenolic acids, flavonoids, and total tannin [58,59]. *C. mollissima* inner shell contains multiple polyphenolic components, which have a beneficial health care function to the human body [60]. Chestnut shells are optimized for alkaline delignification to co-produce ligin and bio-ethanol [61]. Flavonoids procyanidin B3, quercetin-3-O-glycoside, and steroidal sapogenins extracted from *C. mollissima* shells are natural resource pigments, which are natural additives in production in the food industry [62].

### 5.2. Postharvest Biology and Technology

The postharvest quality of chestnut has been a major focus of research in recent years. Chestnuts treated by high-pressure processing (400, 500, and 600 MPa for 5 min at 20 °C) have prolonged storage times, fewer molds and insect larvae, and nuts with higher quality [63]. Chestnuts coated with chitosan have lower abundances of microorganisms under refrigeration (0 °C, 90% HR) after 6 months compared with control chestnuts [64]. Chitosan nanoparticles loaded with thymol coating reduce the decline in the soluble sugar and starch content of chestnut and inhibit the growth of mold and yeast [65].

## 6. Soil, Fertilizer, and Endophytic Fungi

### 6.1. Soil and Fertilizer Conditions

The optimal soil conditions for chestnut are deep, soft, and original volcanic soil rich in phosphorus, potassium, and organic matter [66]. The content of most macronutrients (e.g., total nitrogen, available phosphorus, potassium, and calcium) in chestnut leaves and orchards is low, which indicates that chestnut trees could benefit from fertilization [67]. Nine- and ten-year-old chestnut trees are largest when they are fertilized with suitable amounts of N, P, and K fertilizer [68]. The vertical distribution of plant nutrients under a balanced fertilization regime (N_2_:P_1_:K_2_) is affected by soil type (e.g., loam, clay loam, and sandy loam) [69]. Different levels of N and P application increase the content of some elements, such as N and Cu, but decrease the content of other elements (e.g., Fe and Mn under N application, and K, Ca, and Mg under P application) in the leaves of chestnut [70]. The N concentration in chestnut leaves and the average nut yield are higher under lime application plus a compound NPK fertilizer than lime plus phosphorus fertilizer and unfertilized control orchards [71]. Potassium silicate application increases the accumulation of phytoliths in the leaf, cell wall, and xylem and enhances the tolerance of plants to high temperature by increasing the efficiency of photosystem II and the content of photosynthetic pigments [72]. Regardless of the sampling period, the concentration of boron in leaves is the factor most strongly correlated with chestnut productivity, and leaves are the most useful tissue for the early diagnosis of boron deficiency during the bloom period [73]. The conversion of native forest to Chinese chestnut plantations increases the total nitrogen stock but decreases carbon storage in soil following intensive management [74].

### 6.2. Endophytic Fungi of Chestnut Trees

Trees can obtain their nutrients through mycorrhizae and nutrient recycling. The fitness and productivity of chestnuts are affected by different types of fungi, and ectomycorrhizal fungi (EMF) affect both the roots and above-ground parts of plants [75,76,77]. The symbionts of EMF and chestnut contribute to the strength of the roots, including their ability to deeply penetrate the soil, which enhances the growth of chestnut trees. Various planting methods have been used to increase the growth and survival of EMF to promote EMF root colonization and the restoration of chestnuts [78,79]. The EMF species *Scleroderma* spp., *Laccaria* spp., and *Cenococcum geophilum* can disperse and colonize new habitats rapidly, including young chestnut trees [80], and *Amanita*, *Boletus*, *Cantharellus*, *Cortinarius*, *Lactarius*, *Russula*, and *Tricholoma* are dominant in mature trees [81]. Chestnut blight induces decreases in the rate of photosynthesis, EMF colonization, and species diversity, and blight does not have an effect on the parent trees or neighboring five-month-old seedlings, which are more tolerant of reductions in the content of photosynthate [82]. The success of EMF colonization differs among chestnut tree species infected with *P. cinnamomi* and healthy ones, and EMF have more extrametrical hyphae in healthy trees than in infected trees [83]. The decrease in mycorrhizal root tip density and ECM species richness might explain the defoliation caused by Swiss needle cast disease [84]. Thus, EMF can be used to enhance the survival of chestnut when chestnut trees are reintroduced to regions vulnerable to blight outbreaks.

## 7. Genetic Research with SSRs

SSR-based analyses are used to characterize the genetic diversity and genetic structure of plant populations, as well as patterns of kinship, all of which have conservation benefits [85,86,87]. Trees from southeastern France, Corsica, and other regions show differentiation in 10 SSRs [88]. A total of 17 polymorphic microsatellite primers have been shown to be effective for the cross-species amplification of several Fagaceae members, including Japanese and European chestnut and interspecific hybrids of Chinese and Japanese chestnut [89]. An analysis of 16 SSRs has revealed two genotypes: the major genotype ‘Marroni’ and another genotype observed in different varieties of Italian chestnuts [90]. An analysis of 10 SSR loci and 20 morphological characteristics of 68 chestnut plants in the Piedmont of Italy revealed 36 different genotypes, and four gene pools were shown to contribute to the formation of the population according to analysis of the genetic structure of germplasm [91]. Seven of the SSR markers selected from 146 unique fingerprints were core markers for rapidly distinguishing different accessions of Chinese chestnut resources [92]. Twelve new SSR markers were developed for the identification of 216 chestnut accessions, and 21 synonym groups and 189 different genotypes were identified [88]. A total of 272 chestnut individuals were genotyped with 24 SSR markers, and analysis of five outlier loci showed that they were significantly associated with environmental variables [32].

Genetic diversity significantly differed within and among populations of chestnut in southern Spain, but this pattern was not observed using neutral microsatellite markers [93]. Three species of Chinese chestnut represent a single monophyletic clade according to a phylogeny of *Castanea* based on chloroplast *trn*T-L-F sequence data, and the North American and European species were sister groups [94]. A total of 52 accessions of Swiss chestnut cultivars were separated into groups of different genotypes by clustering analysis with random amplified polymorphic DNA, amplified fragment length polymorphism DNA, and inter-simple sequence repeats [95]. Microsatellite markers and various analysis methods were used to obtain a single overall gene pool with a diverse admixture of genotypes, and the genotypes in Britain and Ireland differed from those in other areas in Europe [96]. An analysis of the genetic differences in four chestnut populations in the western Balkans and northern Italy using 21 microsatellite markers revealed an introgression zone in the northern Balkans that might be established through gene flow from the above regions [97]. *FIR059* exhibited three private alleles for drought-susceptible individuals and two private alleles for drought-tolerant individuals according to an analysis with eight EST-SSR markers; these could thus provide candidate markers for predicting the drought tolerance of *C. sativa* trees [98]. The genotypes of chestnut seeds vary, but the mass of fruit might be an important target of selection [99].

## 8. Ecological Environment of Chestnut Orchards

The ecological features of chestnuts are manifold and include soil conditions, temperature, and altitude; natural and anthropogenic factors are also important, as well as whether management is traditional or intensive [2,66,100,101,102]. Compared with infested chestnut seeds, non-infested and mast seeds are cached more by rodents, which promotes seed germination and forest regeneration [103]. *C. sativa* is an invasive alien species on the Canary Islands that many have suggested should not be completely eradicated given that they are reservoirs of lichen biodiversity [11]. Study of the American chestnut in the southwestern portion of its historical range has revealed that a suitable habitat is present at higher elevations and in regions with higher forest canopy cover [104]. Claudia Mattioni et al. (2017) identified three main gene pools of sweet chestnut and a significant genetic barrier among populations, and this provided new insights into the biogeographic history, geographic locations of different gene pools, and priority areas possessing high genetic diversity in Europe [105]. A 30-year study has shown that the spread of chestnut in northeastern Turkey is mostly driven by temperature and total precipitation [106]. The phenology, morphometric parameters, ripening time, and average yield of different genotypes of sweet chestnut can vary among locations [107]. Compared with other co-occurring tree species, sweet chestnut has a lower survival probability because of its low shade tolerance, poor competitiveness, preference for dry conditions, and summer temperatures [12]. Latitudinal variation among populations is associated with variation in phenology and the xerothermic index; the phenology of chestnut trees in central and southern Mediterranean populations was earlier than that of the northern populations, and drought plays a key role in this pattern [108].

## 9. China Has Abundant Chestnut Germplasm Resources

Asian chestnut varieties, including Chinese and Japanese chestnut, are considered important resources of genotypes resistant to *P. cinnamomi*, *C. parasitica*, and the chestnut gall wasp [30,109,110]. However, few breeders have focused on breeding varieties resistant to these pathogens in China. This is one of the causes of the differences in research on pest control and breeding among Asian and European/American countries. For example, a priority in current research in the Americas, Europe, Australia, and New Zealand is to acquire new varieties with resistance to pathogens, elite genotypes, and tolerance of a wide range of environmental conditions because of the high risk of damage by disease and insects. However, in China, the focus has been on improving the quality of nuts or their use as a source of useful secondary metabolisms. *Dichocrocis punctiferalis* GUENEE (Lepidoptera: Pyralidae) is the most harmful pest of ‘Wukeli’, and this pest has been studied in 15 chestnut orchards from four villages in ‘Luotian’ County (Huanggang City, Hubei Province, China) in 2021 (Figure 3) (data not shown).

Many orchards that have been abandoned due to attacks by pests and disease in American countries are being re-established, and there is a major focus on selecting resistant varieties or hybrids [109,110,111]. China has the most abundant and suitable chestnut resources to support this initiative.

## 10. Conclusions

A diversified strategy based on careful consideration of the different chestnut cultivation regions is needed to increase chestnut production and improve the marketing of chestnut products. The main measures that could be taken include helping farmers with afforestation efforts, strategically planting chestnut orchards, and implementing scientifically based agronomic and chemical/biological measures to combat diseases and pests to increase the productivity and quality of wood and nuts.

## Figures and Tables

**Figure 1 plants-11-02111-f001:**
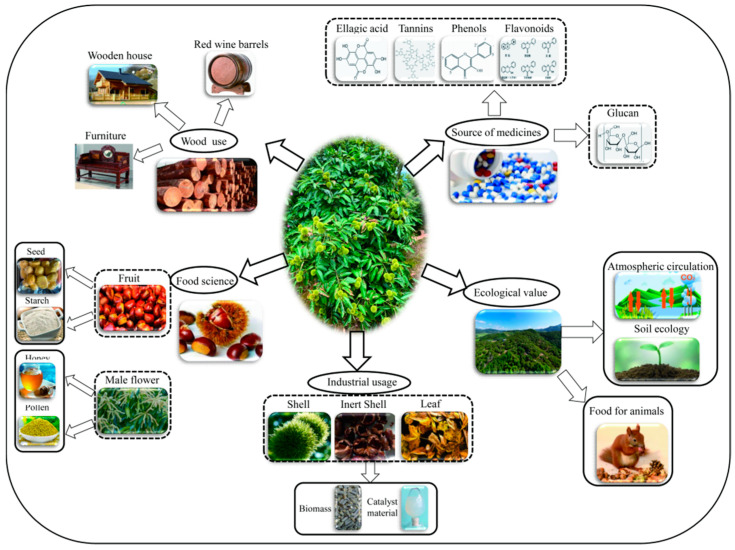
Flowchart of the processing and utilization of the wood, leaves, and nuts of chestnut trees. *Castanea mollissima* Bl. is a multi-value species with high ecological value that is widely used in the food, nutraceutical, pharmaceutical, and industrial raw material industries.

**Figure 2 plants-11-02111-f002:**
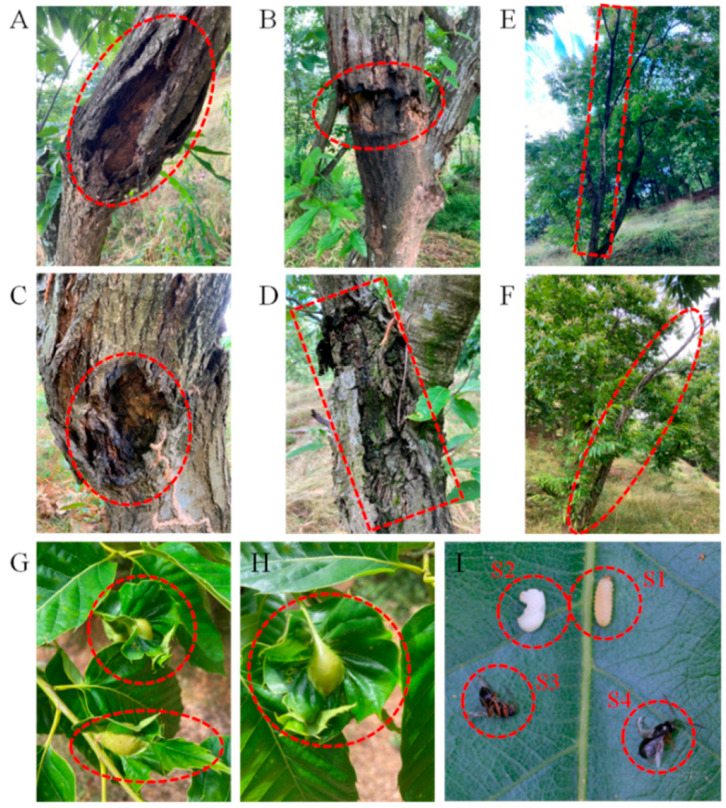
Main diseases and pests of chestnut. Characteristics of chestnut stem after *Cryphonectria parasitica* infection (**A**–**F**). The bark and phloem of chestnut trees can be severely damaged by *Cryphonectria parasitica*; however, the xylem can remain fully intact and functional, and the trees can remain alive (**A**–**D**). However, the xylem can be severely damaged after infection, and the trees can die (**E**,**F**). Leaf galls induced by *Dryocosmus kuriphilus* on a chestnut tree (**G**,**H**); larvae (**I**-S1,S2) and adults (**I**-S3,S4) of *Dryocosmus kuriphilus.* Three different development forms of *Dryocosmus kuriphilus* appear simultaneously on the same chestnut tree in May. The photos were taken in a private orchard in Qingyuan County, Lishui City, Zhejaing Province, China.

**Figure 3 plants-11-02111-f003:**
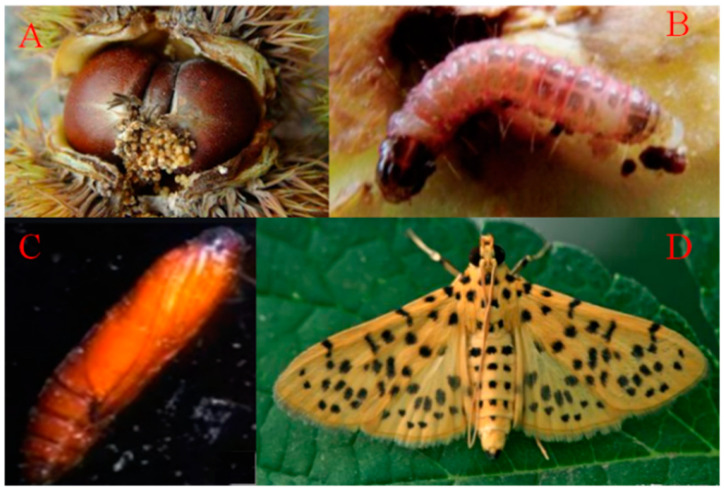
*Dichocrocis punctiferalis* GUENEE is the main insect pest of Chinese chestnut in Luotian County, Hubei Province, China. Chestnuts attacked by *D. punctiferalis* (**A**), and different stages of the life cycle of *D. punctiferalis*: larva (**B**), pupa (**C**), and adult (**D**). Photos were taken in a private orchard in Luotian County, Huanggang City, Hubei Province, China in 2021.

## Data Availability

Not applicable.

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
