# Peer review of "A Review of the Stress Resistance, Molecular Breeding, Health Benefits, Potential Food Products, and Ecological Value of Castanea mollissima"

_plants, 2022, doi:10.3390/plants11162111_

Round 1

Reviewer 1 Report

The manuscript, with the title: "A rewiew of the stress resistance, molecular breeding, healt benefits, potential food products, bioactive compounds, and ecological value of Castanea mollissima" tries to give an overview of the situation of chestnut production and processing. The topic is very diverse and wide-ranging, so even a broad overview can only be sketchy. Each subject area would be suitable for writing an overview article separately.

Since the world importance of the chestnut is indisputable, I recommend that the authors mention the four most important edible chestnut species (C. sativa, C. crenata, C. mollissima, C. dentata) already in the introduction. I also consider it important to mention Oceania (Australia and New Zealand) and Africa (e.g. Tunisia and Canary Islands) among the growing regions of the world.

In the case of ink disease, not only P. cinnamomi is the causative agent, but several other agents have also been described, e.g. P. cambivora and P. citricola, etc.). Referral of these is also necessary for the complete picture.

When discussing the potential use in the health industry, I would like to point out to the authors that not only the shell can be used to extract valuable compounds, but also the buds in the spring.

P. cinnamomic (correctly P. cinnamomi) should be corrected in line 145 of the manuscript.

After the modifications and corrections mentioned above, I recommend publishing the manuscript.

Reviewer 2 Report

Abstract should better represent the review and aim of the manuscript. Please reduce the number of keywords.

Introduction is too poor. Authors should improve this part by adding more information on the Figure 1. It is very interesting but not so much clear. Please add also an image with a higher resolution.

Paragraphs on plant diseases are longer than others. Is this a review only focused on these traits? Otherwise also the other sections should be reported with the same level of deepening.

More information should be reported on the bioactive compounds that may be extracted from Chinese chestnuts and their properties. Otherwise "bioactive compounds" should be deleted from the title. A little paragraph on the main analytical strategies used for chestnut chemical characterization should be added (also as a table).

Conclusions should be reported in order to summarize the main issues of the review. Please delete the redundant information.
